# Large Scale Manifold Balanced Clustering

Fangfang Li [1] [*]   Quanxue Gao [1]   Xingyu Xue [1] [*]

## Abstract

Manifold clustering has demonstrated strong capability in capturing complex data structures and has been widely studied in cluster analysis. However, many existing methods mainly focus on combining K-means with manifold learning, while overlooking the consistency between data structures and clustering labels, and often suffer from high computational cost when handling large scale data. To address these issues, we propose a manifold balanced clustering method based on anchor induced distance(LMBC), grounded in the relationship between K-means clustering and manifold learning. Specifically, the LMBC uses label information to guide the construction of the manifold structure, thereby ensuring consistency between data structures and clustering labels. To enable large scale clustering, we introduce an anchor induced distance representation that models manifold structure in a compact anchor space, significantly reducing computational complexity while preserving essential structural information. Furthermore, to naturally maintain class balance during clustering, we maximize the Schatten-$p$ norm of the label representation and provide theoretical analysis to support its effectiveness. Experimental results on several benchmark datasets demonstrate the effectiveness and scalability of the proposed method.

## 1. Introduction

Clustering is a fundamental unsupervised learning task that aims to partition data into meaningful groups and has been widely studied in machine learning and pattern analysis. Owing to the abundance of unlabeled data in real-world applications, clustering plays an important role in exploratory data analysis and knowledge discovery.

[*]Equal contribution [1]School of Telecommunications Engineering, Xidian University, Shaanxi 710071, China. Correspondence to: Quanxue Gao <qxgao@xidian.edu.cn>.

*Proceedings of the $43^{rd}$ International Conference on Machine Learning*, Seoul, South Korea. PMLR 306, 2026. Copyright 2026 by the author(s).

Among various clustering techniques, distance-based methods such as K-means (Hartigan & Wong, 1979) are particularly popular due to their simplicity and computational efficiency. Despite their success, K-means and its variants (Xin et al., 2023; Liu, 2023; Yao et al., 2021) rely heavily on Euclidean distance and global centroid-based representations, which limits their ability to handle data with complex nonlinear structures. Recent studies have also explored center-free formulations of K-means to alleviate centroid dependence (Nie et al., 2022b; Pei et al., 2023), yet these methods still face challenges in capturing complex data structures. To overcome this limitation, manifold-based clustering methods have been proposed to exploit the intrinsic geometric structure of data by preserving local neighborhood relationships (Souvenir & Pless, 2005; Cai, 2015). These methods are effective in capturing nonlinear data distributions; however, in most existing approaches, manifold construction and clustering assignment are treated as two separate stages, which may lead to inconsistency between the learned data structure and the resulting cluster labels.

To further improve scalability on large-scale datasets, anchor based clustering methods have attracted increasing attention by shifting graph construction and optimization to anchor space (Wang et al., 2024; Nie et al., 2024; Xie et al., 2025b). While such methods significantly reduce computational complexity, many existing approaches either still define clustering objectives implicitly at the sample level or perform clustering on anchor graphs without explicitly considering cluster balance. As a result, these methods may suffer from performance degradation or degenerate solutions when data distributions are highly skewed. Clustering balance has long been recognized as a critical issue in unsupervised learning, aiming to avoid highly imbalanced cluster size distributions. Early studies addressed this problem from a graph partitioning perspective, such as Ratio Cut and Normalized Cut. More recent works introduce balance related regularization terms to softly encourage uniform cluster distributions (Xie et al., 2025a; Li et al., 2025). Although effective to some extent, these balanced clustering strategies are rarely integrated with anchor-level clustering and manifold-aware representations in a unified framework, especially in large-scale settings.

Motivated by these observations, we propose a large-scale manifold balanced clustering(LMBC) method based on

anchor-induced distance. The proposed method performs clustering directly in the anchor space and leverages anchor-induced distances to capture manifold structure. Notably, we provide a new perspective on the Schatten-$p$ norm in clustering. While the Schatten-$p$ norm is conventionally minimized in applications such as image denoising (Gao et al., 2021) and anomaly detection (Du et al., 2023), we show that maximizing the Schatten-$p$ norm naturally promotes balanced cluster assignments. The main contributions of this work can be summarized as follows:

- We propose a large-scale manifold balanced clustering method based on anchor-induced distance, which performs clustering directly in the anchor space and significantly reduces computational complexity while preserving intrinsic manifold structure.

- We explicitly promote the consistency between manifold structure and cluster labels by coupling manifold-aware distances with clustering assignments, leading to more coherent and stable clustering results.

- We establish a principled connection between clustering balance and the Schatten-$p$ norm, and reveal that maximizing the Schatten-$p$ norm naturally encourages balanced cluster partitions, providing a novel perspective on clustering with explicit balance regularization.

## 2. Related Work

### 2.1. Distance-Based and Manifold Clustering

Distance-based clustering methods have been extensively studied due to their simplicity and efficiency. Representative approaches such as K-means (Hartigan & Wong, 1979) aim to minimize intra-cluster variance measured by Euclidean distance. Kernel-based extensions further enhance their ability to handle nonlinear data distributions by mapping samples into high-dimensional feature spaces (Xin et al., 2023; Liu, 2023; Yao et al., 2021). Despite their effectiveness in many scenarios, these methods rely primarily on global distance measures and often struggle to capture complex intrinsic data structures. Manifold-based clustering methods address this limitation by exploiting local neighborhood relationships to model the underlying geometric structure of data. By constructing similarity graphs and preserving local affinities, these approaches can capture nonlinear manifolds embedded in high-dimensional spaces (Souvenir & Pless, 2005; Cai, 2015). Some methods further integrate clustering with manifold learning by jointly optimizing similarity graphs and clustering assignments (Zhang et al., 2023; Nie et al., 2014; Wang et al., 2022). However, in most existing approaches, manifold construction and clustering assignment are still treated as loosely coupled processes, which may lead to inconsistency between the learned manifold

structure and the final clustering results.

### 2.2. Anchor-Based Clustering for Large-Scale Data

To improve scalability on large scale datasets, anchor clustering methods have been proposed to reduce computational and memory costs by representing data using a compact set of anchors. Early anchor based approaches primarily employ anchors to accelerate graph construction or approximate sample level similarity relationships, while the clustering objectives remain defined over samples (Zhao et al., 2024). For example, EDCAG (Wang et al., 2024) alternately optimizes sample level and anchor level labels to approximate graph cut objectives, but explicitly maintains a sample level label matrix. Subsequent studies further move optimization variables to the anchor space. FCAG (Nie et al., 2024) directly learns anchor labels and transfers them to samples through anchor-sample associations, while fuzzy anchor-based methods such as FFCAG (Nie et al., 2022a) and SFCAG (Liu et al., 2023) perform clustering on anchors and propagate membership degrees back to samples. Although these approaches reduce computational complexity, the clustering objectives are still implicitly defined at the sample level, and anchors mainly serve as intermediate representations. More recently, FAGC (Xie et al., 2025b) performs clustering directly on the anchor–anchor graph, making the computational complexity primarily dependent on the number of anchors and thus suitable for large-scale datasets. However, FAGC focuses on maximizing within-cluster similarity on the anchor graph and does not explicitly incorporate cluster balance constraints, which may result in highly imbalanced anchor partitions.

### 2.3. Balanced Clustering

Clustering balance has been recognized as a fundamental problem in unsupervised learning, aiming to avoid degenerate solutions or highly imbalanced cluster size distributions. Early studies addressed this issue from the perspective of graph partitioning, with Ratio Cut (Chan et al., 1994) and Normalized Cut (Shi & Malik, 2000) being representative examples. These methods implicitly encourage balanced partitions by exploiting the spectral properties of the graph Laplacian. Subsequent works introduced hard balanced clustering strategies by explicitly enforcing cluster size constraints during optimization. For instance, Malinen et al. (Malinen & Fränti, 2014) proposed approximately distributing samples evenly across clusters, while FCALS (Xie et al., 2025a) imposed explicit lower bounds on cluster sizes through hard constraints on the label matrix. Although such methods can guarantee balanced cluster cardinalities, they often lack flexibility and increase optimization difficulty. In contrast, soft balanced clustering methods incorporate balance related regularization terms into the objective function. Lin et al. (Lin et al., 2019) explicitly treat cluster sizes as op-

timization variables, while Mi et al. (Mi et al., 2024) encourage uniform cluster distributions by maximizing Shannon entropy. In addition, norm-based regularization strategies promote competition among clusters by penalizing similarity between different columns of the label or representation matrix (Wang et al., 2023; Liu et al., 2017). While effective in many scenarios, most existing balanced clustering methods are designed for sample level clustering and are rarely integrated with anchor level representations or manifold aware clustering frameworks in large scale settings.

# 3. Methodology

## 3.1. Preliminaries and Notations

Let $\mathbf{X} \in \mathbb{R}^{n \times d}$ denote the dataset with $n$ samples and $d$-dimensional features. We select $m$ anchors ($m \ll n$) to represent the data, denoted as $\mathbf{A} \in \mathbb{R}^{m \times d}$. The relationships between data points and anchors are captured by an anchor graph $\mathbf{M} \in \mathbb{R}^{n \times m}$. $\mathbf{W} \in \mathbb{R}^{m \times m}$ denotes the anchor-level distance matrix. $\mathbf{Z} \in \mathbb{R}^{m \times K}$ denotes the anchor label matrix, where $K$ is the number of clusters. The anchors are selected using the variance-based decorrelated anchor (VDA) strategy (Xia et al., 2023).

## 3.2. Manifold K-means in Anchor Space

K-means is a classical distance-based clustering method that assigns labels by minimizing distances between data samples and cluster centers. Despite its efficiency, it mainly relies on global distance information in the original space and has limited capability in capturing complex local data structures. In contrast, manifold learning techniques are effective in modeling local geometric relationships but cannot directly produce clustering labels. Existing methods attempt to combine K-means with manifold learning by jointly incorporating distance information and manifold regularization. However, such combinations usually treat the two components as independent modules and do not explicitly enforce consistency between the learned manifold structure and clustering labels, which may lead to inconsistent assignments.

A natural way to address this issue is to revisit the relationship between K-means and manifold learning from a unified perspective, and consider the K-means as a special case of manifold learning in the original space (Gao et al., 2025). Following this observation, we extend the manifold K-means to the anchor space, as shown in Eq. (1).

$$\min_{\mathbf{Z}} \sum_{i=1}^{m} \sum_{j=1}^{m} \|\mathbf{a}_i - \mathbf{a}_j\|_2^2 \, s_{ij}, \quad \text{s.t. } \mathbf{Z} \in Ind, \quad (1)$$

where $\mathbf{a}_i$ denotes the $i$-th anchor and $\mathbf{Z} \in \mathbb{R}^{m \times K}$ is the anchor label matrix. The label-induced manifold structure is defined as $\mathbf{S} = \mathbf{Q}\mathbf{Q}^{\top}$, with $\mathbf{Q} = \mathbf{Z}\mathbf{P}^{-1/2}$ and $\mathbf{P} = \text{diag}(\sum_{i=1}^{m} z_{i1}, \sum_{i=1}^{m} z_{i2}, \dots, \sum_{i=1}^{m} z_{iK})$.

In this formulation, the anchor labels $\mathbf{Z}$ directly guide the learning of the manifold structure $\mathbf{S}$, and clustering is performed on the learned manifold, ensuring label consistency among anchors lying on the same manifold surface.

To facilitate optimization, the pairwise distance objective in Eq. (1) can be equivalently rewritten in a matrix trace form.

$$\min_{\mathbf{Z}} \text{tr}(\mathbf{Q}^T \mathbf{W} \mathbf{Q}) = \min_{\mathbf{Z}} \text{tr}(\mathbf{Z}^T \mathbf{W} \mathbf{Z} \mathbf{P}^{-1}), \quad (2)$$

where $\mathbf{W} \in \mathbb{R}^{m \times m}$ denotes the anchor-level distance matrix.

## 3.3. Manifold–Label Consistency Modeling

Although Eq. (2) provides a compact trace formulation that intrinsically couples the manifold structure with clustering labels, it does not prevent degenerate or highly imbalanced clustering solutions. To address this issue, we introduce Theorem 3.1, which establishes a connection between the Schatten-$p$ norm of the label matrix and clustering balance.

**Theorem 3.1.** *Given $m_1 + \dots + m_K = m$, the maximum value of Eq.(3) is reached when $m_1 = \dots = m_K = \frac{m}{K}$. In this case, $\mathbf{Z}$ is discrete and exhibits a balanced class distribution.*

$$\max_{\mathbf{Z} \geq 0, \mathbf{Z}\mathbf{1}=\mathbf{1}} \|\mathbf{Z}\|_{sp}^p \quad (3)$$

*Proof.* Let $\sigma_j(\mathbf{Z})$ denote the $j$-th largest singular value of $\mathbf{Z}$, and $\tau_j(\mathbf{Z}^T\mathbf{Z})$ denote $j$-th largest eigenvalue of $\mathbf{Z}^T\mathbf{Z}$.

$$\|\mathbf{Z}\|_{sp}^p = \sum_{j=1}^{K} \sigma_j^p(\mathbf{Z}) = \sum_{j=1}^{K} (\tau_j(\mathbf{Z}^T\mathbf{Z}))^{\frac{p}{2}} = \sum_{j=1}^{K} \delta_j^{\frac{p}{2}} \quad (4)$$

where $\delta_j = \tau_j(\mathbf{Z}^T\mathbf{Z})$.

Let $\boldsymbol{\delta} = [\delta_1, \dots, \delta_K]^T \in \mathbb{R}^{K \times 1}$, $\beta_1 = \dots = \beta_K = \frac{1}{K}$. When $0 < p \leq 2$, $f(\delta_j) = \delta_j^{\frac{p}{2}}$ is a concave function with respect to $\delta_j$, then according to Jensen inequality, we have

$$f\left(\sum_{j=1}^{K} \beta_j \delta_j\right) \geqslant \sum_{j=1}^{K} \beta_j f(\delta_j) \quad (5)$$

Equality holds if and only if $\delta_1 = \delta_2 = \dots = \delta_K$.

For the inequality(5), by simplifying the right side of the inequality, we obtain

$$\sum_{j=1}^{K} \beta_j f(\delta_j) = \frac{1}{K} \sum_{j=1}^{K} f(\delta_j) = \frac{1}{K} \|\mathbf{Z}\|_{sp}^p \quad (6)$$

Similarly, by simplifying the left side of the inequality(5), we obtain

$$\sum_{j=1}^{K} \beta_j \delta_j = \sum_{j=1}^{K} \beta_j \tau_j(\mathbf{Z}^T\mathbf{Z}) = \frac{1}{K} \sum_{j=1}^{K} \tau_j(\mathbf{Z}^T\mathbf{Z}) = \frac{1}{K} \|\mathbf{Z}\|_F^2 \quad (7)$$

Further, $f(\sum_{j=1}^{K} \beta_j \delta_j) = f(\frac{1}{K}\|\mathbf{Z}\|_F^2) = (\frac{1}{K}\|\mathbf{Z}\|_F^2)^{\frac{p}{2}}$.

Therefore, we can transform the original inequality(5) into

$$K(\frac{1}{K}\|\mathbf{Z}\|_F^2)^{\frac{p}{2}} \geqslant \|\mathbf{Z}\|_{sp}^p \qquad (8)$$

Equality holds if and only if $\delta_1 = \delta_2 = \ldots = \delta_K$. This indicates that, under the constraint $\delta_1 = \ldots = \delta_K$, maximizing $\|\mathbf{Z}\|_{sp}^p$ is equivalent to maximizing $K(\frac{1}{K}\|\mathbf{Z}\|_F^2)^{\frac{p}{2}}$, i.e

$$\max \|\mathbf{Z}\|_{sp}^p \Leftrightarrow \max \|\mathbf{Z}\|_F^2 = \max \sum_i \sum_j z_{ij}^2$$
$$s.t. z_{ij} \geq 0, \sum_j z_{ij} = \mathbf{1}, \delta_1 = \delta_2 = \ldots = \delta_K \qquad (9)$$

In Eq.(9), each row of $\mathbf{Z}$ is independent, so for each row of $\mathbf{Z}$, Eq.(9) becomes $\max \sum_{j=1}^{K} z_{ij}^2$. The maximum of Eq. (9) is achieved when $\mathbf{z}_i$ contains only one element equal to 1 and all remaining elements are 0, where the maximum value is equal to 1. Therefore, the objective $(\|\mathbf{Z}\|_F^2)^{\frac{p}{2}}$ attains its maximum only when $\mathbf{Z}$ is a discrete matrix. In this case, $\mathbf{Q} = \mathbf{Z}^T\mathbf{Z}$ becomes a diagonal matrix, whose $j$-th diagonal entry corresponds to the eigenvalue $\delta_j$, which is also equal to the anchor number $m_j$ of the $j$-th cluster.

Noting that $\sum_{j=1}^{K} m_j = m$, we have

$$\frac{1}{K}\sum_{j=1}^{K} \delta_j = \frac{1}{K}\sum_{j=1}^{K} m_j = \frac{m}{K} \qquad (10)$$

According to the equality condition of Eq. (5), we have $\delta_1 = \delta_2 = \cdots = \delta_K$. Combining with $\delta_j = m_j$ and $\sum_{j=1}^{K} m_j = m$, we further obtain

$$\delta_1 = \delta_2 = \cdots = \delta_K = m_1 = m_2 = \cdots = m_K = \frac{m}{K} \quad (11)$$

Therefore, $\|\mathbf{Z}\|_{sp}^p$ reaches its maximum when $m_1 = m_2 = \cdots = m_K = \frac{m}{K}$. □

Theorem 3.1 shows that maximizing the Schatten-$p$ norm in Eq. (3) leads to an approximately balanced partition. When the optimal solution is achieved, it holds that $(\mathbf{Z}^T\mathbf{Z})^{1/2} = \frac{m}{K}\mathbf{I}$. Motivated by this result, we reformulate the objective in Eq. (2) into the following continuous optimization model:

$$\min_{\mathbf{Z} \geq 0, \mathbf{Z1}=\mathbf{1}} \text{tr}(\mathbf{Z}^T\mathbf{W}\mathbf{Z}) - \alpha\|\mathbf{Z}\|_{sp}^p. \qquad (12)$$

Under the optimal solution of model (12), the label matrix $\mathbf{Z}$ becomes discrete and yields approximately balanced clusters. Here, $\mathbf{W}$ denotes a general anchor-level distance matrix, whose specific construction will be introduced later.

## 3.4. Optimization

Existing optimization methods mainly focus on the minimization of the Schatten $p$-norm, while the maximization problem in Eq. (12) is difficult to solve directly. Therefore, we adopt a first-order Taylor approximation strategy to iteratively optimize the objective.

Specifically, let $f(\mathbf{Z}) = \|\mathbf{Z}\|_{sp}^p$. At the $t$-th iteration, the first-order Taylor expansion of $f(\mathbf{Z})$ at $\mathbf{Z}^{(t)}$ is given by

$$f(\mathbf{Z}) = f(\mathbf{Z}^{(t)}) + <\nabla f(\mathbf{Z}^{(t)}), \mathbf{Z} - \mathbf{Z}^{(t)}> \qquad (13)$$

where $\mathbf{Z}^{(t)}$ denotes the solution at the $t$ iteration, $\nabla f(\mathbf{Z}^{(t)})$ is the derivative of $\|\mathbf{Z}\|_{sp}^p$, as given in Theorem 3.2.

**Theorem 3.2.** *The derivative of $\|\mathbf{Z}\|_{sp}^p$ with respect to $\mathbf{Z}$ is:*

$$\mathbf{F} = \frac{\partial\|\mathbf{Z}\|_{sp}^p}{\partial\mathbf{Z}} = p\mathbf{U}\mathbf{\Sigma}^{-1}|\mathbf{\Sigma}|^p\mathbf{V}^T \qquad (14)$$

*where $\mathbf{Z} = \mathbf{U}\mathbf{\Sigma}\mathbf{V}^T$, $\mathbf{\Sigma}^{-1}$ is the Moore-Penrose pseudo-inverse of $\mathbf{\Sigma}$. See Appendix for specific proof.*

Ignoring the constant terms in Eq. (13), Eq. (12) can be iteratively solved as

$$\mathbf{Z}^{(t+1)} = \underset{\mathbf{Z1}=\mathbf{1},\mathbf{Z}\geqslant\mathbf{0}}{\text{argmin}} \ \text{tr}(\mathbf{Z}^T\mathbf{W}\mathbf{Z}) - \alpha <\nabla f(\mathbf{Z}^{(t)}), \mathbf{Z}>$$
$$= \min_{\mathbf{Z1}=\mathbf{1},\mathbf{Z}\geqslant\mathbf{0}} tr(\mathbf{Z}^T\mathbf{W}\mathbf{Z}) - \alpha tr(\mathbf{F}^T\mathbf{Z}) \qquad (15)$$

where $\mathbf{F} = \nabla f(\mathbf{Z}^{(t)}) = \frac{\partial\|\mathbf{Z}\|_{sp}^p}{\partial\mathbf{Z}}\Big|_{\mathbf{Z}^{(t)}}$. At each iteration, $\mathbf{F}$ is updated according to the current solution $\mathbf{Z}^{(t)}$.

Let $\mathbf{Z} = \begin{bmatrix} \mathbf{z}^i \\ \mathbf{Z}_0 \end{bmatrix}$, and $\mathbf{W} = \begin{bmatrix} w_{ii} & \mathbf{w}_{i0}^T \\ \mathbf{w}_{i0} & \mathbf{W}_0 \end{bmatrix}$, where $\mathbf{Z}_0 \in \mathbb{R}^{(N-1)\times K}$, $\mathbf{w}_{i0} \in \mathbb{R}^{(N-1)\times 1}$, $\mathbf{W}_0 \in \mathbb{R}^{(N-1)\times(N-1)}$.

We have

$$\mathbf{Z}^T\mathbf{W}\mathbf{Z} = \begin{bmatrix} (\mathbf{z}^i)^T & (\mathbf{Z}_0)^T \end{bmatrix} \begin{bmatrix} w_{ii} & \mathbf{w}_{i0}^T \\ \mathbf{w}_{i0} & \mathbf{W}_0 \end{bmatrix} \begin{bmatrix} \mathbf{z}^i \\ \mathbf{Z}_0 \end{bmatrix}$$
$$= (\mathbf{z}^i)^T w_{ii}\mathbf{z}^i + (\mathbf{Z}_0)^T\mathbf{w}_{i0}\mathbf{z}^i \qquad (16)$$
$$+ (\mathbf{z}^i)^T\mathbf{w}_{i0}^T\mathbf{Z}_0 + (\mathbf{Z}_0)^T\mathbf{W}_0\mathbf{Z}_0$$

Let $\mathbf{F} = \begin{bmatrix} \mathbf{f}^i \\ \mathbf{F}_0 \end{bmatrix}$, then $\mathbf{F}^T\mathbf{Z} = (\mathbf{f}^i)^T\mathbf{z}^i + (\mathbf{F}_0)^T\mathbf{Z}_0$

$$\mathbf{Z}^T\mathbf{W}\mathbf{Z} - \alpha\mathbf{F}^T\mathbf{Z}$$
$$= (\mathbf{z}^i)^T w_{ii}\mathbf{z}^i + (\mathbf{Z}_0)^T\mathbf{w}_{i0}\mathbf{z}^i + (\mathbf{z}^i)^T\mathbf{w}_{i0}^T\mathbf{Z}_0 \qquad (17)$$
$$+ (\mathbf{Z}_0)^T\mathbf{W}_0\mathbf{Z}_0 - \alpha(\mathbf{f}^i)^T\mathbf{z}^i - \alpha(\mathbf{F}_0)^T\mathbf{Z}_0$$

Through the properties of the trace operation, we derive

$$\text{tr}(\mathbf{Z}^T\mathbf{W}\mathbf{Z} - \alpha\mathbf{F}^T\mathbf{Z})$$
$$= \text{tr}((\mathbf{z}^i)^T w_{ii}\mathbf{z}^i + 2\mathbf{z}^i\mathbf{Z}_0^T\mathbf{w}_{i0} - \alpha\mathbf{z}^i(\mathbf{f}^i)^T)$$
$$= \mathbf{z}^i(\mathbf{z}^i)^T w_{ii} + \mathbf{z}^i(2\mathbf{Z}_0^T\mathbf{w}_{i0} - \alpha(\mathbf{f}^i)^T) \qquad (18)$$
$$= \mathbf{z}^i(\mathbf{z}^i)^T w_{ii} + \mathbf{z}^i\mathbf{h}$$

where $\mathbf{h} = 2\mathbf{Z}_0^{\mathrm{T}}\mathbf{w}_{i0} - \alpha(\mathbf{f}^i)^{\mathrm{T}}$.

Thus, the problem of updating the i-th row of $\mathbf{Z}$ can be:

$$\min_{\mathbf{z}^i \mathbf{1} = 1} \mathbf{z}^i \mathbf{h} \tag{19}$$

Given $w_{ii} = 0$ for $i = 1, 2, \ldots, N$, the solution for $\mathbf{z}^i$ is

$$z_{ib} = \begin{cases} 1, b = \arg\min_{j} (2\mathbf{Z}^{\mathrm{T}}\mathbf{w}_i - \alpha(\mathbf{f}^i)^{\mathrm{T}})_j \\ 0, \text{otherwise.} \end{cases} \tag{20}$$

When updating the $i$-th row of $\mathbf{Z}$, computing $\mathbf{Z}^\top \mathbf{w}_i$ requires $\mathcal{O}(mK)$ time. Updating all rows of $\mathbf{Z}$ therefore leads to a per-iteration complexity of $\mathcal{O}(m^2 K)$.

**Acceleration strategy.** To reduce the computational cost, we precompute and store $\mathbf{Z}^\top \mathbf{w}_i$ for all $i$, which requires $\mathcal{O}(m^2)$ time. With this caching strategy, updating each row of $\mathbf{Z}$ only involves $\mathcal{O}(K)$ operations.

Moreover, since each row of $\mathbf{Z}$ contains only one nonzero entry, updating the $i$-th row only affects two anchor indices. Accordingly, the cached values can be efficiently updated via simple incremental operations:

$$\mathbf{z}_e^\top \mathbf{w}_i \leftarrow \mathbf{z}_e^\top \mathbf{w}_i - w_{ie}, \qquad \mathbf{z}_b^\top \mathbf{w}_i \leftarrow \mathbf{z}_b^\top \mathbf{w}_i + w_{ib}. \tag{21}$$

As a result, the per-iteration time complexity is reduced to $\mathcal{O}(mK)$, which significantly improves the efficiency of the proposed optimization algorithm. Algorithm 1 summarizes the procedure for solving model (15), and Algorithm 2 presents the overall optimization framework.

---

**Algorithm 1** Optimizing $\mathbf{Z}$

1: **Input** distance matrix $\mathbf{W} \in \mathbb{R}^{m \times m}$, matrix $\mathbf{F}$, hyperparameter $\alpha$.
2: **Initialize** label matrix $\mathbf{Z} \in \mathbb{R}^{m \times K}$
3: **repeat**
4:     Calculate and store $\mathbf{Z}^{\mathrm{T}}\mathbf{W}$, $N_k = \mathbf{z}_k^{\mathrm{T}}\mathbf{1}$
5:     **while** $\mathbf{Z}$ not converge **do**
6:       for each $i \in [1, N]$
7:       $e \leftarrow$ the index of element 1 in $\mathbf{z}^i$
8:       **if** $N_e = 1$ **then**
9:         continue
10:       **end if**
11:       $b \leftarrow$ obtained via (20)
12:       **if** $b \neq e$ **then**
13:         Update $\mathbf{z}_k^{\mathrm{T}}\mathbf{w}_i (k = e, b)$ via (21)
14:       **end if**
15:     **end while**
16: **until** convergence
17: **Output** $\mathbf{Z} \in \mathbb{R}^{m \times K}$

---

The output $\mathbf{Z}$ corresponds to anchor labels. The labels of data points are obtained via $\mathbf{AZ}$, where each sample

---

**Algorithm 2** solve objective function (12)

1: **Input** distance matrix $\mathbf{W} \in \mathbb{R}^{m \times m}$, cluster number $K$, hyperparameter $\alpha$.
2: **Initialize** label matrix $\mathbf{Z} \in \mathbb{R}^{m \times K}$
3: **repeat**
4:     update matrix $\mathbf{F}$ by Eq. (25);
5:     update matrix $\mathbf{Z}$ by Algorithm 1;
6: **until** convergence
7: **Output** $\mathbf{Z} \in \mathbb{R}^{m \times K}$

---

is assigned to the cluster with the maximum value in its corresponding row.

### 3.5. Anchor-Induced Distance Construction

Instead of directly constructing a sample-level pairwise distance matrix, we propose an anchor-level distance representation to improve scalability while preserving structural information. The key idea is to exploit anchors as compact structural proxies and to derive a discriminative distance representation through a nonlinear transformation.

We first establish an anchor graph $\mathbf{M} \in \mathbb{R}^{n \times m}$ ($m \ll n$), which encodes the local relationships between samples and anchors(Xia et al., 2023)(Nie et al., 2016; Liu et al., 2010). Each row $\mathbf{m}_i$ is obtained by solving

$$\min_{\mathbf{m}_i^\top \mathbf{1} = 1, \ \mathbf{m}_i \geq 0} \sum_{j=1}^{\theta} m_{ij} \|\mathbf{x}_i - \mathbf{a}_j\|_2^2 + \gamma \|\mathbf{m}_i\|_2^2, \tag{22}$$

where $\theta$ denotes the number of neighboring anchors. This formulation encourages each sample to be represented by a small number of nearby anchors(Nie et al., 2016; Liu et al., 2010).

Thus, the anchor-level structural association matrix is constructed as(Ma et al., 2025)

$$\mathbf{B} = \mathbf{M}^\top \Delta^{-1} \mathbf{M} \tag{23}$$

where $\Delta \in \mathbb{R}^{n \times n}$ is a diagonal matrix with $\Delta_{ii} = \sum_{j=1}^{m} m_{ij}$. The matrix $\Delta$ is introduced to balance the contributions of different samples in aggregating anchor-level structural relationships. In practice, a small constant $\varepsilon$ can be added to the diagonal of $\Delta$ when necessary to ensure numerical stability.

We then transform $\mathbf{B}$ into an anchor-level distance matrix $\mathbf{W} \in \mathbb{R}^{m \times m}$ via a nonlinear mapping:

$$w_{ij} = \frac{2}{1 + (2\pi b_{ij})^2} \tag{24}$$

By operating on the anchor-level distance matrix $\mathbf{W}$, the proposed framework avoids the quadratic complexity of sample-wise distance computation and provides an efficient distance surrogate for subsequent clustering.

### 3.6. Time and Space Complexity Analysis

**Time Complexity:** Our approach needs to update $\mathbf{Z}$ and $\mathbf{F}$ iteratively. (1) For $\mathbf{F}$, it needs $\mathcal{O}(mK^2)$ by Eq. (25). (2) For $\mathbf{Z}$, refer to the previous optimization acceleration section, it needs to precompute and store $\mathbf{Z}^{\mathrm{T}}\mathbf{w}_i (i = 1, 2, \ldots, m)$ once in advance, which needs $\mathcal{O}(m^2K)$. Then, during the iteration of $\mathbf{Z}$, it just needs $\mathcal{O}(mKt)$, where $t$ is the number of iterations of Algorithm 1, $K$ and $m$ represent the number of clusters and anchors, respectively. Therefore, the whole time complexity of Algorithm 2 is $\mathcal{O}((mK^2 + mKt)T + m^2K)$, where, $T$ is the number of iterations of Algorithm 2.

**Space Complexity:** The storage memories for $\mathbf{W}$, $\mathbf{F}$ and $\mathbf{Z}$ need $\mathcal{O}(m^2)$, $\mathcal{O}(mK)$, and $\mathcal{O}(mK)$, respectively. The overall space complexity of our approach is $\mathcal{O}(m^2)$.

## 4. Experiments

### 4.1. Datasets

We conduct exhaustive experiments on eight benchmark datasets: AR (Martinez & Benavente, 1998), gisette(Guyon & Dror, 2004), isolet(Cole & Fanty, 1991), USPS (Hull, 1994), Pendigits [1], cifar10, MNIST (Lecun et al., 1998), and Covtype(Chang & Lin, 2011). In Table 1, we provide a detailed overview of these datasets.

*Table 1.* The information of datasets.

|          | AR    | gisette | isolet | USPS  |
|----------|-------|---------|--------|-------|
| Samples  | 3,120 | 7,000   | 7,797  | 9,298 |
| Features | 2,000 | 5,000   | 617    | 256   |
| Classes  | 120   | 2       | 26     | 10    |

|          | Pendigits | cifar10 | MNIST  | Covtype |
|----------|-----------|---------|--------|---------|
| Samples  | 10,992    | 50,000  | 70,000 | 581,012 |
| Features | 16        | 256     | 784    | 54      |
| Classes  | 10        | 10      | 10     | 7       |

### 4.2. Results

Table 2 and Table 3 reports the clustering performance of all compared methods in terms of ACC, NMI and Purity on different benchmark datasets. Overall, the proposed method consistently achieves the best or second-best performance across most datasets and evaluation metrics, demonstrating its strong clustering capability and robustness.

We first compare the proposed method with representative sample-level clustering algorithms, including K-means, KKM (Tzortzis & Likas, 2008), CDKM (Nie et al., 2022b), K-sum (Pei et al., 2023), ERCAN (Wang et al., 2022) and RKM (Lin et al., 2019). As shown in Table 2 and Table 3, the proposed method significantly outperforms these meth-

ods on most datasets. This is mainly because these approaches operate directly in the original sample space and rely on global distance measures, which limits their ability to capture complex manifold structures and makes them less scalable to large-scale datasets. Moreover, several sample-level methods such as ERCAN, KKM and K-SUM require constructing a full $N \times N$ distance or similarity matrix, which easily leads to out-of-memory (OM) issues on large dataset. In contrast, the proposed method operates on an anchor-level distance matrix of size $M \times M$ with $M \ll N$, significantly reducing both memory consumption and computational overhead.

We further compare the proposed method with anchor-based clustering approaches including LCSOG (Zhang et al., 2023), FCAG (Xie et al., 2025b), ACLR (Ma et al., 2025) and FCALS-D (Xie et al., 2025a). Although these methods reduce computational complexity by introducing anchors, most of them still treat anchors mainly as intermediate representations for label propagation or approximation, and do not explicitly enforce consistency between manifold structure and clustering labels. For example, FCAG and ACLR mainly focus on learning anchor labels and propagating them to samples, while LCSOG jointly optimizes similarity graphs and clustering assignments but does not explicitly consider clustering balance. On the Gisette dataset, LCSOG exhibits a highly imbalanced clustering result, where only one sample is assigned to one cluster while almost all remaining samples are grouped into the other cluster. This indicates that even anchor-level methods may collapse into trivial solutions without proper balance regularization. In contrast, on naturally imbalanced datasets such as AR and Covtype, LMBC still achieves competitive performance without enforcing artificial cluster balance, indicating its ability to adapt to the underlying data distribution.

We also compare the proposed method with representative landmark-based manifold learning approaches, including roseland (Shen & Wu, 2020) and Landmark Isomap (L-Isomap) (Silva & Tenenbaum, 2002). These methods first learn low-dimensional manifold embeddings using landmarks or reference points and then perform K-means clustering in the embedding space. Although they improve scalability by avoiding the construction of full pairwise affinity matrices, they mainly focus on manifold preservation rather than directly optimizing clustering objectives. As shown in Table 2 and Table 3, their clustering performance is generally inferior to the proposed method.

Our method performs clustering directly in the anchor space and explicitly integrates manifold structure with label assignment. Furthermore, by maximizing the Schatten-$p$ norm of the anchor label matrix, the proposed method provides a soft and theoretically grounded mechanism to encourage balanced partitions without imposing hard constraints. This

---

[1]https://odds.cs.stonybrook.edu/pendigits-dataset/

*Table 2.* The clustering performances on the AR, isolet, gisette, and USPS datasets.

| Datasets | AR | | | isolet | | | gisette | | | USPS | | |
|---|---|---|---|---|---|---|---|---|---|---|---|---|
| Methods | ACC | NMI | Purity | ACC | NMI | Purity | ACC | NMI | Purity | ACC | NMI | Purity |
| $K$-means | 0.2514 | 0.5574 | 0.2749 | 0.5469 | 0.7154 | 0.5958 | 0.6843 | 0.1167 | 0.6843 | 0.6458 | 0.6026 | 0.7129 |
| KKM | 0.2529 | 0.5346 | 0.2715 | 0.5238 | 0.7029 | 0.5621 | 0.5029 | 0.0000 | 0.5029 | 0.6872 | 0.6437 | 0.7565 |
| L-Isomap | 0.2803 | 0.5818 | 0.2983 | 0.5313 | 0.7104 | 0.5808 | 0.8379 | 0.4027 | 0.8379 | 0.7571 | 0.7593 | 0.8317 |
| CDKM | 0.2653 | 0.5700 | 0.2862 | 0.5328 | 0.7159 | 0.5837 | 0.6854 | 0.1180 | 0.6854 | 0.6526 | 0.6094 | 0.7237 |
| K-sum | 0.3002 | 0.6004 | 0.3132 | 0.6269 | 0.7347 | 0.6402 | 0.6516 | 0.0676 | 0.6516 | 0.6802 | 0.6274 | 0.7486 |
| ERCAN | 0.3936 | 0.5949 | 0.4295 | 0.5509 | 0.7521 | 0.6048 | 0.9421 | 0.6839 | 0.9421 | 0.7185 | 0.7801 | 0.7929 |
| RKM | 0.2641 | 0.5752 | 0.3215 | 0.6299 | 0.7346 | 0.6387 | 0.7066 | 0.1269 | 0.7066 | 0.6241 | 0.5748 | 0.7003 |
| LCSOG | 0.3423 | 0.4992 | 0.3817 | 0.5352 | 0.6678 | 0.5529 | 0.5017 | 0.0013 | 0.5017 | 0.7448 | 0.7778 | 0.7922 |
| FCAG | 0.3592 | 0.6656 | 0.3787 | 0.5833 | 0.7241 | 0.6078 | 0.8714 | 0.4554 | 0.8714 | 0.7895 | 0.7048 | 0.8073 |
| ACLR | 0.3272 | 0.5376 | 0.3497 | 0.5639 | 0.7566 | 0.6260 | 0.9131 | 0.5779 | 0.9131 | 0.6794 | **0.7892** | 0.7949 |
| FCALS-D | 0.2548 | 0.5102 | 0.3290 | 0.5893 | 0.7116 | 0.6463 | 0.8359 | 0.3962 | 0.8359 | 0.6974 | 0.6843 | 0.7573 |
| LMBC | **0.4212** | **0.7005** | **0.4417** | **0.6597** | **0.7731** | **0.6655** | **0.9679** | **0.7954** | **0.9679** | **0.8560** | 0.7661 | **0.8560** |

*Table 3.* Clustering performance on four large-scale datasets. "–" denotes excessive computational time, and "OM" denotes out-of-memory.

| Datasets | Pendigits | | | cifar10 | | | MNIST | | | Covtype | | |
|---|---|---|---|---|---|---|---|---|---|---|---|---|
| Methods | ACC | NMI | Purity | ACC | NMI | Purity | ACC | NMI | Purity | ACC | NMI | Purity |
| $K$-means | 0.6963 | 0.6705 | 0.7260 | 0.7608 | 0.7776 | 0.8105 | 0.5436 | 0.4950 | 0.5900 | 0.2506 | 0.0617 | -0.0045 |
| KKM | 0.7859 | 0.7139 | 0.7859 | 0.3613 | 0.2379 | 0.3794 | 0.4900 | 0.4551 | 0.5420 | OM | OM | OM |
| L-Isomap | 0.7364 | 0.7036 | 0.7541 | 0.8966 | 0.8225 | 0.8966 | 0.6051 | 0.5701 | 0.6465 | – | – | – |
| CDKM | 0.7027 | 0.6697 | 0.7226 | 0.8964 | 0.8375 | 0.8980 | 0.5442 | 0.4949 | 0.5887 | 0.2506 | 0.0617 | 0.4905 |
| K-sum | 0.7562 | 0.6743 | 0.7562 | 0.9200 | 0.8475 | 0.9200 | 0.5485 | 0.4605 | 0.5714 | OM | OM | OM |
| ERCAN | 0.7921 | 0.8011 | 0.7996 | – | – | – | – | – | – | OM | OM | OM |
| RKM | 0.7296 | 0.6639 | 0.7296 | 0.9227 | 0.8528 | 0.9227 | 0.5529 | 0.4825 | 0.5903 | OM | OM | OM |
| LCSOG | 0.8432 | 0.7871 | 0.8432 | 0.8405 | 0.8015 | 0.8418 | 0.5754 | 0.5964 | 0.6249 | 0.4191 | 0.0230 | 0.4904 |
| FCAG | 0.7652 | 0.7293 | 0.7817 | 0.9075 | 0.8382 | 0.9075 | 0.6481 | 0.6114 | 0.6791 | 0.2195 | 0.0601 | 0.0162 |
| ACLR | 0.8417 | 0.7908 | 0.8417 | 0.8722 | 0.7994 | 0.8722 | 0.7490 | 0.7286 | 0.7765 | 0.3674 | **0.0965** | 0.0413 |
| FCALS-D | 0.8418 | 0.7748 | 0.8418 | 0.8961 | 0.8246 | 0.8961 | 0.6864 | 0.6915 | 0.7538 | 0.3660 | 0.0491 | **0.5199** |
| LMBC | **0.8594** | **0.8038** | **0.8594** | **0.9267** | **0.8635** | **0.9267** | **0.7964** | **0.7384** | **0.7964** | **0.4230** | 0.0459 | 0.4995 |

makes it particularly suitable for large-scale clustering with both structural consistency and balanced cluster distributions.

### 4.3. Parameters setting and analysis

We first analyze the influence of the anchor ratio on clustering performance, as shown in Figure. 1. The anchor ratio is varied from 0.1 to 1.0. According to the experimental results, the optimal anchor ratios are 0.9 on AR, 0.8 on gisette, 0.26 on isolet, and 0.24 on USPS. These variations are related to dataset characteristics. AR contains a relatively large number of classes and high feature dimensionality, which requires a higher anchor ratio to adequately capture the fine-grained manifold structure. Similarly, gisette is a high dimensional dataset, and a larger anchor ratio helps preserve discriminative structural information. In contrast, isolet and USPS have relatively lower feature dimensionality and fewer classes, and their data distributions exhibit stronger redundancy. For these datasets, a smaller anchor ratio is sufficient to represent the essential manifold structure while significantly reducing computational cost. In practice, the anchor ratio should be selected according to dataset characteristics. For datasets with high feature di-

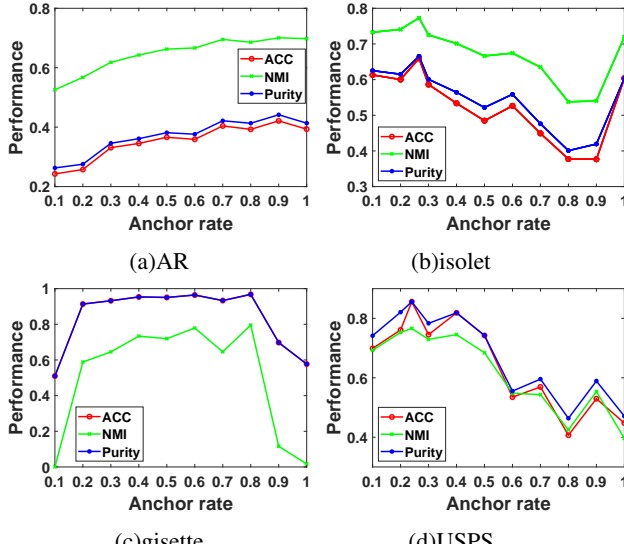

(a)AR  (b)isolet

(c)gisette  (d)USPS

*Figure 1.* Effect of anchor ratio.

mensionality, a relatively higher anchor ratio is preferred to capture fine-grained manifold structures, while for datasets with lower dimensionality or more redundant structures, a

smaller anchor ratio is usually sufficient. However, increasing the anchor ratio also raises computational cost and may reduce the scalability advantage of the proposed method. Therefore, the anchor ratio should be chosen to balance representation capability and computational efficiency.

Figure 2 reports the clustering accuracy under different combinations of the Schatten $p$-norm parameter $p$ and its weight $\lambda$. Specifically, a grid search strategy is adopted, where $p$ varies from 1.0 to 1.9 with a step size of 0.1, and $\lambda$ ranges from 0.1 to 1.0 with a step size of 0.1. The results show that the proposed method achieves competitive performance under a wide range of parameter settings. For different datasets, the optimal performance is obtained at different $(p, \lambda)$ combinations, while no abrupt performance degradation is observed. This indicates that the proposed method is not sensitive to the choice of $p$ and $\lambda$, and stable clustering results can be achieved through simple grid based parameter tuning.

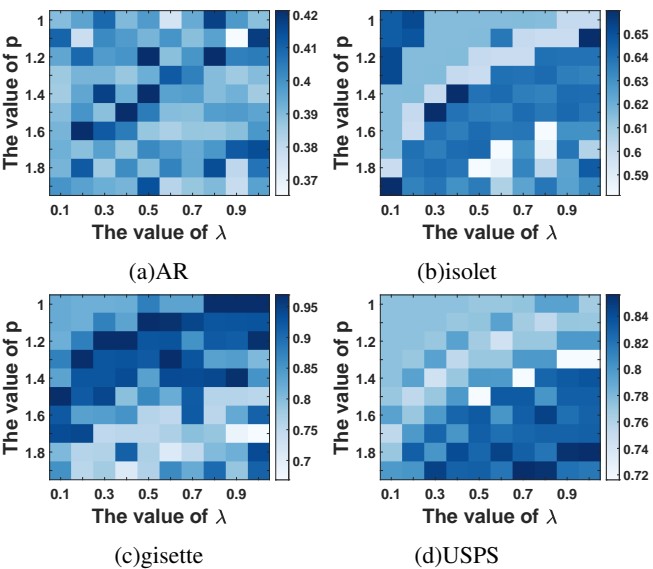

(a)AR        (b)isolet

(c)gisette        (d)USPS

*Figure 2.* Effect of parameter $\lambda$ and $p$.

### 4.4. Ablation Experiment

To evaluate the contribution of each component, we conduct an ablation study by removing one module at a time: (1) w/o Bal., removing the Schatten-p regularization term; (2) w/o Anch., replacing the anchor-induced distance with Euclidean distance between anchors; and (3) w/o Mani., replacing the manifold consistency term with a k-means clustering objective.

As shown in Table 4, removing any component leads to consistent performance degradation across datasets. Specifically, excluding the Schatten-p regularization reduces clustering quality, indicating its effectiveness in enhancing representation robustness. Replacing the anchor-induced distance

with Euclidean anchor distance causes the largest decline, highlighting the importance of anchor structure in capturing meaningful relationships. In addition, removing the manifold consistency term also negatively affects performance, showing its role in preserving intrinsic data structure. Overall, these results confirm that all components contribute complementarily to the effectiveness of the proposed framework.

### 4.5. Clustering Balance Analysis on Synthetic Datasets

To evaluate the balancing capability of our algorithm, we synthetically constructed Balanced-8 (B8) dataset. The B8 dataset consists of 8 categories, also with 100 samples per category. The experimental results are illustrated in Figure 3. By comparing the clustering distributions of the K-means, LCSOG, and LMBC algorithms on the B8 dataset, clear performance differences can be observed. Figure 3(a) shows the clustering results of K-means, which reveal a severe class imbalance: class 5 contains only 34 samples, while class 1 dominates with 175 samples. This imbalance is mainly attributed to K-means optimizing solely the Euclidean distance without considering cluster balance, and its sensitivity to initialization often leads to uncontrolled cluster sizes. Figure 3(c) presents the clustering distribution of LCSOG, with cluster sizes of 101, 104, 99, 98, 97, 107, 100, and 94. Compared with K-means, LCSOG produces a substantially more balanced distribution, although slight variations among clusters still remain. In sharp contrast, Figure 3(e) presents the clustering distribution of the LMBC algorithm, where all classes are evenly distributed with 100±3 samples each. This balance is achieved through our proposed Schatten-p norm regularization mechanism, which promotes class balance by maximizing the Schatten-p norm of the label matrix.

And we constructed an imbalanced dataset, Imbalanced-8 (ImB8), which consists of 8 categories with varying sample sizes: 50, 120, 60, 150, 100, 200, 80 and 40. Using K-means, we obtained an ACC of 0.8175. The clustering distribution is visualized in Figure 3(b). The resulting cluster sizes were 53, 104, 99, 130, 95, 149, 91 and 79, showing significant deviation from the original class distribution. For the anchor graph clustering method LCSOG, the ACC improved to 0.9325. As shown in Figure 3(d), the cluster sizes were 53, 122, 78, 150, 99, 197, 99, and 2. Although its ACC was higher than that of K-means, the clustering distribution still showed clear distortion from the original imbalanced class proportions. In contrast, our LMBC algorithm achieved a significantly higher ACC of 0.9825. The clustering distribution is shown in Figure 3(f), where the cluster sizes were 53, 121, 59, 149, 99, 198, 79, and 42. These results show that in imbalanced datasets, our method maintains the clustering relationships of closely spaced samples, naturally preserving the original imbalanced distribution rather than

*Table 4.* Ablation study results of different model variants.

| Datasets | AR | | | isolet | | | gisette | | | USPS | | |
|---|---|---|---|---|---|---|---|---|---|---|---|---|
| Methods | ACC | NMI | Purity | ACC | NMI | Purity | ACC | NMI | Purity | ACC | NMI | Purity |
| w/o Bal. | 0.3894 | 0.6871 | 0.4099 | 0.6264 | 0.7496 | 0.6373 | 0.8213 | 0.3228 | 0.8213 | 0.7729 | 0.7368 | 0.7729 |
| w/o Anch. | 0.3356 | 0.6202 | 0.3551 | 0.5625 | 0.6441 | 0.5660 | 0.6080 | 0.0455 | 0.6080 | 0.6516 | 0.5667 | 0.6627 |
| w/o Mani. | 0.3590 | 0.6508 | 0.3782 | 0.5616 | 0.6745 | 0.5702 | 0.7471 | 0.1843 | 0.7471 | 0.7316 | 0.6389 | 0.7638 |
| LMBC | **0.4212** | **0.7005** | **0.4417** | **0.6597** | **0.7731** | **0.6655** | **0.9679** | **0.7954** | **0.9679** | **0.8560** | **0.7661** | **0.8560** |

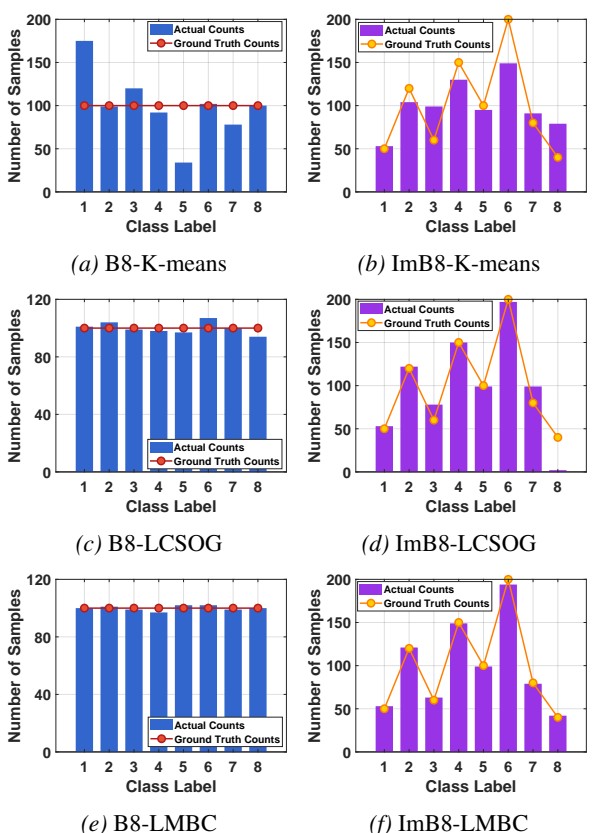

*Figure 3.* Visualization of clustering distributions on B8 v.s. ImB8 Datasets.

forcing balance. Overall, our method achieves reasonable clustering results under different data conditions, achieving cluster balance on balanced datasets while preserving the original distribution of data on imbalanced datasets.

### 4.6. Convergence

We evaluate the convergence and clustering performance of our optimization algorithm. Specifically, we analyze the convergence of the $\mathbf{Z}$-update in Algorithm 1, and the results are presented in Figure 4. As shown in Figure 4(a)(c), the error between consecutive iterations, $\mathbf{Z}_t$ and $\mathbf{Z}_{t-1}$, decreases rapidly as the number of iterations increases, indicating stable convergence of Algorithm 1. Moreover, Figure 4(b)(d) shows that the objective value of Algorithm 2 decreases

monotonically and converges within a few iterations, which can be attributed to the effective optimization of $\mathbf{Z}$ during the iterative process.

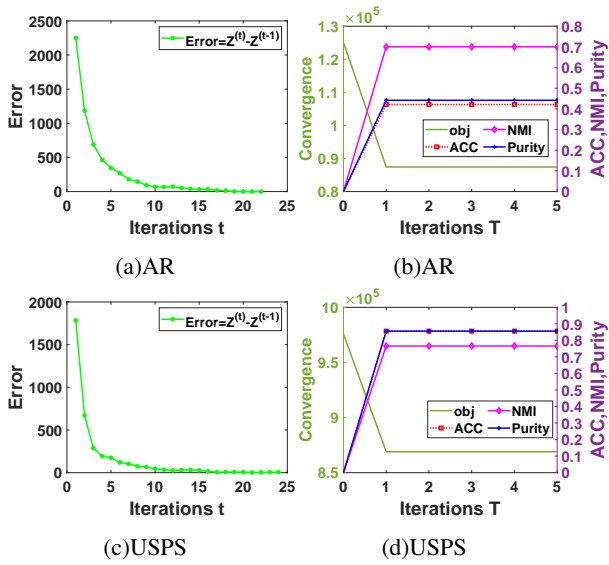

*Figure 4.* (a)(c) show the convergence of $\mathbf{Z}$, and (b)(d) show the objective value and clustering performance.

## 5. Conclusion

This paper proposes a manifold balanced clustering method based on an anchor-induced distance. By constructing the manifold structure from clustering labels and optimizing labels directly on the learned manifold, the proposed method ensures consistency between manifold structure and clustering assignments. Moreover, class balance is naturally encouraged by maximizing the Schatten-$p$ norm of the label representation, which effectively avoids degenerate solutions. Extensive experimental results on benchmark datasets demonstrate the effectiveness, robustness, and scalability of the proposed method.

## Acknowledgements

This work was supported by the National Natural Science Foundation of China, Grant No. 625B2137 and 62576263; the Natural Science Basic Research Program of Shaanxi

Province, Grant No. 2025JC-QYCX-051; Shaanxi Fundamental Science Research Project for Mathematics and Physics, Grant No. 25JSZ008; the Fundamental Research Funds for the Central Universities and the Innovation Fund of Xidian University.

## Impact Statement

This paper aims to advance the field of machine learning, and we do not identify any specific ethical or societal concerns requiring special emphasis.

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

# 6. Appendix

## 6.1. Visualization of distance matrix

JAFFE is a facial expression dataset containing 213 images with 676 features and 10 categories (Lyons et al., 1998).

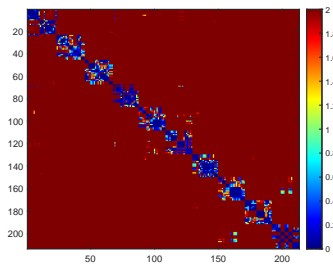

*Figure 5.* Visualization of the anchor-induced distance matrix on the JAFFE dataset. Color from blue to red represents distance from small to large.

## 6.2. Proof of the Theorem 3.2

**Theorem 3.2.** The derivative of $\|\mathbf{Z}\|_{sp}^p$ with respect to $\mathbf{Z}$ is:

$$\mathbf{F} = \frac{\partial \|\mathbf{Z}\|_{sp}^p}{\partial \mathbf{Z}} = p\mathbf{U}\boldsymbol{\Sigma}^{-1}|\boldsymbol{\Sigma}|^p\mathbf{V}^{\mathrm{T}} \tag{25}$$

where $\mathbf{Z} = \mathbf{U}\boldsymbol{\Sigma}\mathbf{V}^{\mathrm{T}}$, $\boldsymbol{\Sigma}^{-1}$ is the Moore-Penrose pseudo-inverse of $\boldsymbol{\Sigma}$.

*Proof.* Let $\|\mathbf{Z}\|_{sp}^p = \sum_{i=1} \sigma_i^p(\mathbf{Z})$ be the p power of the Schatten p-norm of matrix $\mathbf{Z}$, where $\sigma_i(\mathbf{Z})$ denote the $i$-th largest singular value of $\mathbf{Z}$. We express the singular values in terms of the eigenvalues of $\mathbf{Z}^T\mathbf{Z}$. Specifically,

$$\sigma_i(\mathbf{Z}) = \tau_i\sqrt{\mathbf{Z}^T\mathbf{Z}} = \tau_i(\mathbf{A}^{\frac{1}{2}}) \tag{26}$$

where $\tau_i(\mathbf{Z}^T\mathbf{Z})$ denote $i$-th largest eigenvalue of $\mathbf{Z}^T\mathbf{Z}$, $\mathbf{A} = \mathbf{Z}^T\mathbf{Z}$.

Thus, the $p$-th power of the singular values can be expressed as:

$$\sigma_i^p(\mathbf{Z}) = \tau_i(\mathbf{A}^{\frac{p}{2}}) \tag{27}$$

The Schatten p-norm of $\mathbf{Z}$ is then:

$$\begin{aligned}\|\mathbf{Z}\|_{sp}^p &= tr(\mathbf{A}^{\frac{p}{2}}) = tr((\mathbf{Z}^T\mathbf{Z})^{\frac{p}{2}}) \\ &= tr((\mathbf{V}^T\mathbf{V}\boldsymbol{\Sigma}^2)^{\frac{p}{2}}) = tr(|\boldsymbol{\Sigma}|^p)\end{aligned} \tag{28}$$

where $\boldsymbol{\Sigma}$ denotes the diagonal matrix of singular values.

Next, the gradient of the Schatten p-norm with respect to $\mathbf{Z}$ is given by:

$$\frac{\partial \|\mathbf{Z}\|_{sp}^p}{\partial \mathbf{Z}} = \frac{\partial tr(|\boldsymbol{\Sigma}|^p)}{\partial \mathbf{Z}} = \frac{tr(\partial |\boldsymbol{\Sigma}|^p)}{\partial \mathbf{Z}} \tag{29}$$

The subdifferential of $\mathbf{Z}$ is:

$$\frac{\partial |\boldsymbol{\Sigma}|}{\partial \mathbf{Z}} = |\boldsymbol{\Sigma}|\,\boldsymbol{\Sigma}^{-1}\frac{\partial \boldsymbol{\Sigma}}{\partial \mathbf{Z}} \tag{30}$$

then,

$$\frac{\partial |\boldsymbol{\Sigma}|^p}{\partial \mathbf{Z}} = p|\boldsymbol{\Sigma}|^{p-1}\frac{\partial |\boldsymbol{\Sigma}|}{\partial \mathbf{Z}} = \frac{p|\boldsymbol{\Sigma}|^{p-1}\partial |\boldsymbol{\Sigma}|}{\partial \mathbf{Z}} \tag{31}$$

Substituting Eq. (30) and Eq. (31) into Eq. (29), we have

$$\frac{\partial \|\mathbf{Z}\|_{sp}^p}{\partial \mathbf{Z}} = \frac{tr(p|\boldsymbol{\Sigma}|^{p-1}|\boldsymbol{\Sigma}|\,\boldsymbol{\Sigma}^{-1}\partial \boldsymbol{\Sigma})}{\partial \mathbf{Z}} \tag{32}$$

According to $\mathbf{Z} = \mathbf{U}\boldsymbol{\Sigma}\mathbf{V}^T$, then

$$\partial \mathbf{Z} = \partial \mathbf{U}\boldsymbol{\Sigma}\mathbf{V}^T + \mathbf{U}\partial \boldsymbol{\Sigma}\mathbf{V}^T + \mathbf{U}\boldsymbol{\Sigma}\partial \mathbf{V}^T \tag{33}$$

By simple algebraic manipulation, we have

$$\mathbf{U}\partial \boldsymbol{\Sigma}\mathbf{V}^T = \partial \mathbf{Z} - \partial \mathbf{U}\boldsymbol{\Sigma}\mathbf{V}^T - \mathbf{U}\boldsymbol{\Sigma}\partial \mathbf{V}^T \tag{34}$$

By multiplying both sides of Eq. (34) on the left by $\mathbf{U}^T$ and on the right by $\mathbf{V}$, we obtain,

$$\partial \boldsymbol{\Sigma} = \mathbf{U}^T\partial \mathbf{Z}\mathbf{V} - \mathbf{U}^T\partial \mathbf{U}\boldsymbol{\Sigma} - \boldsymbol{\Sigma}\partial \mathbf{V}^T\mathbf{V} \tag{35}$$

Furthermore, from the orthogonality of $\mathbf{U}$, we have

$$0 = \partial \mathbf{I} = \partial(\mathbf{U}^T\mathbf{U}) = \partial \mathbf{U}^T\mathbf{U} + \mathbf{U}^T\partial \mathbf{U} \tag{36}$$

Using Eq. (36), the trace of the second term in Eq. (35) can be written as

$$\begin{aligned}tr(\mathbf{U}^T\partial \mathbf{U}\boldsymbol{\Sigma}) &= tr(\boldsymbol{\Sigma}^T\partial \mathbf{U}^T\mathbf{U}) \\ &= -tr(\boldsymbol{\Sigma}\mathbf{U}^T\partial \mathbf{U}) \\ &= -tr(\mathbf{U}^T\partial \mathbf{U}\boldsymbol{\Sigma}) \\ &= 0\end{aligned} \tag{37}$$

Similarly, the trace of the third term in Eq. (35) satisfies

$$tr(\boldsymbol{\Sigma}\partial \mathbf{V}^T\mathbf{V}) = 0 \tag{38}$$

Combining Eqs. (35)–(38), we obtain

$$tr(\partial \boldsymbol{\Sigma}) = tr(\mathbf{U}^T\partial \mathbf{Z}\mathbf{V}) \tag{39}$$

Substituting Eq. (39) into Eq. (32), we have

$$\begin{aligned}\frac{\partial \|\mathbf{Z}\|_{sp}^p}{\partial \mathbf{Z}} &= \frac{tr(p|\boldsymbol{\Sigma}|^{p-1}|\boldsymbol{\Sigma}|\,\boldsymbol{\Sigma}^{-1}\mathbf{U}^T\partial \mathbf{Z}\mathbf{V})}{\partial \mathbf{Z}} \\ &= \frac{tr(\mathbf{V}p|\boldsymbol{\Sigma}|^{p-1}|\boldsymbol{\Sigma}|\,\boldsymbol{\Sigma}^{-1}\mathbf{U}^T\partial \mathbf{Z})}{\partial \mathbf{Z}} \\ &= (\mathbf{V}p|\boldsymbol{\Sigma}|^{p-1}|\boldsymbol{\Sigma}|\,\boldsymbol{\Sigma}^{-1}\mathbf{U}^T)^T \\ &= (\mathbf{V}p|\boldsymbol{\Sigma}|^p\,\boldsymbol{\Sigma}^{-1}\mathbf{U}^T)^T \\ &= p\mathbf{U}\boldsymbol{\Sigma}^{-1}|\boldsymbol{\Sigma}|^p\,\mathbf{V}^T\end{aligned} \tag{40}$$

$\square$

