# Large Scale Manifold Balanced Clustering

## 1. Appendix

### 1.1. Visualization of distance matrix

To illustrate the characteristics of the proposed anchor-induced distance, we visualize and compare it with the Euclidean distance on the JAFFE dataset, as shown in Figure 1. Although the anchor-induced distance is constructed at the anchor level, a sample-level distance matrix is plotted for visualization. As can be observed, the anchor-induced distance yields smaller distances within the same class and larger distances across different classes than the Euclidean distance.

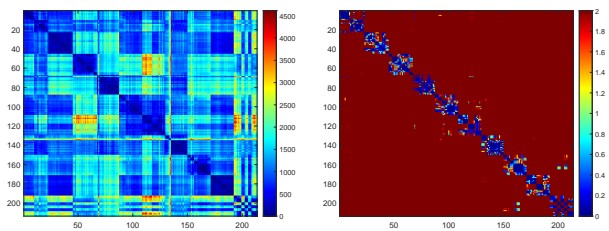

(a)Euclidean distance on JAFFE  (b) anchor-induced distance on JAFFE

*Figure 1.* Visualization of distance matrix $\mathbf{W}$. Color from blue to red represents distance size from small to large.

### 1.2. Proof of the Theorem 3.2

**Theorem 3.2.** The derivative of $\|\mathbf{Z}\|_{sp}^p$ with respect to $\mathbf{Z}$ is:

$$\mathbf{F} = \frac{\partial \|\mathbf{Z}\|_{sp}^p}{\partial \mathbf{Z}} = p\mathbf{U}\boldsymbol{\Sigma}^{-1}|\boldsymbol{\Sigma}|^p\mathbf{V}^{\mathrm{T}} \tag{1}$$

where $\mathbf{Z} = \mathbf{U}\boldsymbol{\Sigma}\mathbf{V}^{\mathrm{T}}$, $\boldsymbol{\Sigma}^{-1}$ is the Moore-Penrose pseudo-inverse of $\boldsymbol{\Sigma}$.

*Proof.* Let $\|\mathbf{Z}\|_{sp}^p = \sum_{i=1} \sigma_i^p(\mathbf{Z})$ be the p power of the Schatten p-norm of matrix $\mathbf{Z}$, where $\sigma_i(\mathbf{Z})$ denote the $i$-th largest singular value of $\mathbf{Z}$. We express the singular values

[1]Anonymous Institution, Anonymous City, Anonymous Region, Anonymous Country. Correspondence to: Anonymous Author <anon.email@domain.com>.

Preliminary work. Under review by the International Conference on Machine Learning (ICML). Do not distribute.

in terms of the eigenvalues of $\mathbf{Z}^T\mathbf{Z}$. Specifically,

$$\sigma_i(\mathbf{Z}) = \tau_i\sqrt{\mathbf{Z}^T\mathbf{Z}} = \tau_i(\mathbf{A}^{\frac{1}{2}}) \tag{2}$$

where $\tau_i(\mathbf{Z}^T\mathbf{Z})$ denote $i$-th largest eigenvalue of $\mathbf{Z}^T\mathbf{Z}$, $\mathbf{A} = \mathbf{Z}^T\mathbf{Z}$.

Thus, the $p$-th power of the singular values can be expressed as:

$$\sigma_i^p(\mathbf{Z}) = \tau_i(\mathbf{A}^{\frac{p}{2}}) \tag{3}$$

The Schatten p-norm of $\mathbf{Z}$ is then:

$$\begin{aligned}\|\mathbf{Z}\|_{sp}^p &= tr(\mathbf{A}^{\frac{p}{2}}) = tr((\mathbf{Z}^T\mathbf{Z})^{\frac{p}{2}}) \\ &= tr((\mathbf{V}^T\mathbf{V}\boldsymbol{\Sigma}^2)^{\frac{p}{2}}) = tr(|\boldsymbol{\Sigma}|^p)\end{aligned} \tag{4}$$

where $\boldsymbol{\Sigma}$ denotes the diagonal matrix of singular values.

Next, the gradient of the Schatten p-norm with respect to $\mathbf{Z}$ is given by:

$$\frac{\partial \|\mathbf{Z}\|_{sp}^p}{\partial \mathbf{Z}} = \frac{\partial tr(|\boldsymbol{\Sigma}|^p)}{\partial \mathbf{Z}} = \frac{tr(\partial |\boldsymbol{\Sigma}|^p)}{\partial \mathbf{Z}} \tag{5}$$

The subdifferential of $\mathbf{Z}$ is:

$$\frac{\partial |\boldsymbol{\Sigma}|}{\partial \mathbf{Z}} = |\boldsymbol{\Sigma}|\boldsymbol{\Sigma}^{-1}\frac{\partial \boldsymbol{\Sigma}}{\partial \mathbf{Z}} \tag{6}$$

and thus, the gradient of $|\boldsymbol{\Sigma}|^p$ with respect to $\mathbf{Z}$ is:

$$\frac{\partial |\boldsymbol{\Sigma}|^p}{\partial \mathbf{Z}} = p|\boldsymbol{\Sigma}|^{p-1}\frac{\partial |\boldsymbol{\Sigma}|}{\partial \mathbf{Z}} = \frac{p|\boldsymbol{\Sigma}|^{p-1}\partial |\boldsymbol{\Sigma}|}{\partial \mathbf{Z}} \tag{7}$$

Substituting Eq. (6) and Eq. (7) back into Eq. (5)

$$\frac{\partial \|\mathbf{Z}\|_{sp}^p}{\partial \mathbf{Z}} = \frac{tr(p|\boldsymbol{\Sigma}|^{p-1}|\boldsymbol{\Sigma}|\boldsymbol{\Sigma}^{-1}\partial \boldsymbol{\Sigma})}{\partial \mathbf{Z}} \tag{8}$$

According to $\mathbf{Z} = \mathbf{U}\boldsymbol{\Sigma}\mathbf{V}^T$,

$$\partial \mathbf{Z} = \partial \mathbf{U}\boldsymbol{\Sigma}\mathbf{V}^T + \mathbf{U}\partial \boldsymbol{\Sigma}\mathbf{V}^T + \mathbf{U}\boldsymbol{\Sigma}\partial \mathbf{V}^T \tag{9}$$

Rearranging the second term:

$$\mathbf{U}\partial \boldsymbol{\Sigma}\mathbf{V}^T = \partial \mathbf{Z} - \partial \mathbf{U}\boldsymbol{\Sigma}\mathbf{V}^T - \mathbf{U}\boldsymbol{\Sigma}\partial \mathbf{V}^T \tag{10}$$

Next, we multiply both sides of the equation by $\mathbf{U}^T$ and $\mathbf{V}$ to obtain:

$$\mathbf{U}^T\mathbf{U}\partial\boldsymbol{\Sigma}\mathbf{V}^T\mathbf{V} = \mathbf{U}^T\partial\mathbf{Z}\mathbf{V} - \mathbf{U}^T\partial\mathbf{U}\boldsymbol{\Sigma}\mathbf{V}^T\mathbf{V} \\ - \mathbf{U}^T\mathbf{U}\boldsymbol{\Sigma}\partial\mathbf{V}^T\mathbf{V} \tag{11}$$

This simplifies to:

$$\partial\boldsymbol{\Sigma} = \mathbf{U}^T\partial\mathbf{Z}\mathbf{V} - \mathbf{U}^T\partial\mathbf{U}\boldsymbol{\Sigma} - \boldsymbol{\Sigma}\partial\mathbf{V}^T\mathbf{V} \tag{12}$$

Consider the identity:

$$0 = \partial\mathbf{I} = \partial(\mathbf{U}^T\mathbf{U}) = \partial\mathbf{U}^T\mathbf{U} + \mathbf{U}^T\partial\mathbf{U} \tag{13}$$

From this, we can get:

$$\begin{aligned} tr(\mathbf{U}^T\partial\mathbf{U}\boldsymbol{\Sigma}) &= tr((\mathbf{U}^T\partial\mathbf{U}\boldsymbol{\Sigma})^T) \\ &= tr(\boldsymbol{\Sigma}^T\partial\mathbf{U}^T\mathbf{U}) \\ &= -tr(\boldsymbol{\Sigma}\mathbf{U}^T\partial\mathbf{U}) \\ &= -tr(\mathbf{U}^T\partial\mathbf{U}\boldsymbol{\Sigma}) \\ &= -tr(\mathbf{U}^T\partial\mathbf{U}\boldsymbol{\Sigma}) \\ &= 0 \end{aligned} \tag{14}$$

In a similar way,

$$tr(\boldsymbol{\Sigma}\partial\mathbf{V}^T\mathbf{V}) = 0 \tag{15}$$

Combine Eq. (12) to Eq. (15), we have

$$tr(\partial\boldsymbol{\Sigma}) = tr(\mathbf{U}^T\partial\mathbf{Z}\mathbf{V}) \tag{16}$$

Substituting Eq. (16) back into Eq. (8),

$$\begin{aligned} \frac{\partial\|\mathbf{Z}\|_{sp}^p}{\partial\mathbf{Z}} &= \frac{tr(p\,|\boldsymbol{\Sigma}|^{p-1}\,|\boldsymbol{\Sigma}|\,\boldsymbol{\Sigma}^{-1}\mathbf{U}^T\partial\mathbf{Z}\mathbf{V})}{\partial\mathbf{Z}} \\ &= \frac{tr(\mathbf{V}p\,|\boldsymbol{\Sigma}|^{p-1}\,|\boldsymbol{\Sigma}|\,\boldsymbol{\Sigma}^{-1}\mathbf{U}^T\partial\mathbf{Z})}{\partial\mathbf{Z}} \\ &= (\mathbf{V}p\,|\boldsymbol{\Sigma}|^{p-1}\,|\boldsymbol{\Sigma}|\,\boldsymbol{\Sigma}^{-1}\mathbf{U}^T)^T \\ &= (\mathbf{V}p\,|\boldsymbol{\Sigma}|^p\,\boldsymbol{\Sigma}^{-1}\mathbf{U}^T)^T \\ &= p\mathbf{U}\boldsymbol{\Sigma}^{-1}\,|\boldsymbol{\Sigma}|^p\,\mathbf{V}^T \end{aligned} \tag{17}$$

$\square$