# OpenReview forum: "Large Scale Manifold Balanced Clustering"
_ICML.cc/2026/Conference — ICML 2026 regular_

### Official Review · Reviewer_SK6Y · 2026-02-20

**Soundness:** 3
**Presentation:** 3
**Significance:** 3
**Originality:** 3
**Overall Recommendation:** 4
**Confidence:** 3

**Summary:**

Existing clustering methods face at least two challenges: balancing cluster labels and scalability. This paper introduces an anchor-based clustering method that simultaneously learns the manifold structure while clustering. The method is scalable (O(M^2K) and produces balanced labels, while still matching the empirical label distribution in a simulated, imbalanced dataset.

**Compliance With Llm Reviewing Policy:**

Affirmed.

**Final Justification:**

The authors’ rebuttal clarifies most technical points and provides additional evidence supporting the method’s scalability and robustness to imbalance. However, the lack of systematic evaluation on varying imbalance levels and more thorough empirical validation of scalability still limits the strength of the evidence, so I will maintain my score.

**Key Questions For Authors:**

How does this work differentiate from Li & Gao (2025) (see arXiv link above) on Schatten-p norm for manifold clustering?

Fig. 3 provides a single, imbalanced synthetic example. How does performance scale with increasing imbalance ratio? Can you provide results on naturally imbalanced benchmarks?

How are anchors chosen?

What criteria were used for dataset/method selection?

**Limitations:**

Limitations and societal impacts are not addressed.

**Strengths And Weaknesses:**

The paper does address a gap in clustering and manifold learning. That is, it provides an example of a manifold-aware, balanced anchor-level clustering suitable for large-scale data.

I reviewed the proof in the main paper, although not entirely thoroughly. It seems to check out with some minor points: The proof states f(\delta_j) = \delta_j^(p/2) is convex for 0 < p ≤ 2, but for p < 2, this function is concave. How might this affect any further points?

Theorem 3.1 is spelled out in https://arxiv.org/abs/2504.20390. Is this paper extending this work? A citation is required.

The paper discusses scalability as one of its main points, but does not empirically validate this point. How is the author's implementation?

Most of the datasets (8 publicly available) are quite balanced, so one would assume that a method that enforces balance in the clustering might do well here. The paper would be strengthened by introducing further (real) imbalanced datasets. Also, the authors should justify the choice of datasets used for validation. Why so few?

Modern manifold learning methods might be introduced (e.g., t-SNE, UMAP, PHATE). How might these be tied in with the anchor-based approach?

---

> ### Author Rebuttal · Authors · 2026-03-31
>
> $\mathbf{Q1}$: We are grateful to the reviewers for pointing out this related study. This arXiv paper performs clustering in the original sample space using pairwise distances, resulting in $\mathcal{O}(N^2)$ complexity. In contrast, our method operates in the anchor space and defines the objective based on anchor-level distances, reducing the complexity to $\mathcal{O}(m^2)$ ($m \ll N$). This makes our approach more suitable for large-scale scenarios. We will incorporate the citations in the revised manuscript.
>
> $\mathbf{Q2}$: Thank you for the insightful suggestion. Figure 3 shows the performance of the proposed method under both balanced and imbalanced settings. While we did not explicitly vary the imbalance ratio in multiple levels, the results demonstrate that the method does not enforce artificial balance and can adapt to the underlying data distribution. In addition, we have evaluated our method on naturally imbalanced datasets such as AR and Covtype. Our method still achieves competitive performance compared to baseline approaches, indicating its robustness in real-world scenarios. We agree that a more systematic analysis of varying imbalance ratios would be valuable, and we will include additional discussion in the revised manuscript.
>
> $\mathbf{Q3}$: Anchors are selected using the variance-based de-correlation anchor selection (VDA) strategy as described in “Tensorized Bipartite Graph Learning for Multi-View Clustering”. This method aims to select representative and diverse anchors by reducing redundancy and preserving the intrinsic data distribution. Specifically, the anchor ratio is selected from the range [0.1, 1.0]. We will clarify this in the revised manuscript.
>
> $\mathbf{Q4}$: Thank you for the valuable comment. We would like to clarify that the datasets used in our experiments include both relatively balanced and imbalanced cases. For example, AR and Covtype exhibit imbalanced class distributions, while the remaining datasets are relatively balanced. These datasets are widely used benchmarks in clustering research and cover diverse characteristics, including varying sample sizes, feature dimensions, and class distributions. Therefore, they provide a comprehensive evaluation of the proposed method under different conditions.
>
> The compared methods include representative sample-level clustering algorithms as well as several anchor-based clustering methods. These methods are relatively recent and closely related to our work. This selection ensures a comprehensive comparison across both traditional clustering frameworks and anchor-based large-scale approaches, covering classical baselines as well as state-of-the-art methods.
>
> $\mathbf{W1}$: We thank the reviewer for the careful reading. We confirm that there is a typographical error in the manuscript. For $0 < p \leq 2$ ,  $f(\delta_j) = \delta_j^{p/2}$  is concave function with respest to $\delta_j$. Although this is a typo, the subsequent derivations remain valid. We will correct this error in the revised manuscript.
>
> $\mathbf{W2}$: We thank the reviewer for pointing out this related work. The arXiv paper studies clustering in the original sample space with $\mathcal{O}(N^2)$ complexity. In contrast, our method operates in the anchor space and reduces the complexity to $\mathcal{O}(m^2)$ ($m \ll N$), making it more suitable for large-scale scenarios. We will include the citation to clarify the relationship.
>
> $\mathbf{W3}$: Thank you for the valuable comment. The scalability of the proposed method mainly comes from the anchor-based representation, which reduces both time and memory complexity from $\mathcal{O}(N^2)$ to $\mathcal{O}(m^2)$, where $m \ll N$. This allows our method to handle large-scale datasets where many sample-based methods become impractical due to high computational cost. To further validate this, we will include runtime comparisons in the revised manuscript. Preliminary results show that our method achieves competitive or lower runtime (e.g., 4.19s on CIFAR-10, 6.82s on MNIST, and 271.18s on Covtype), while several baseline methods encounter out-of-memory (OM) issues on large datasets such as Covtype. These results empirically support the scalability advantage of the proposed method.
>
> $\mathbf{W5}$: Thank you for the insightful suggestion. In our work, the term “manifold” refers to the structural information learned from clustering labels within our optimization framework, rather than explicit manifold embedding methods such as t-SNE, UMAP, or PHATE. These methods are primarily designed for nonlinear dimensionality reduction and visualization, rather than direct clustering. In principle, they can be incorporated as a preprocessing step to obtain low-dimensional representations, which can then be used as input to our anchor-based clustering framework. Exploring a tighter integration between such manifold learning techniques and anchor-based clustering is an interesting direction, which we leave for future work.

---

> > ### Author Rebuttal · Reviewer_SK6Y · 2026-04-01
> >
> > More systemic imbalanced problems and empirical validation of scalability would still be helpful.

---

> > > ### Author Response · Authors · 2026-04-02
> > >
> > > We sincerely thank the reviewer for the constructive suggestions.
> > >
> > > (1)	To provide a more systematic evaluation of imbalance robustness, we further construct additional datasets based on B8 with varying imbalance ratios (IR = 1, 5, 10, 100), where $IR = \max(n_i) / \min(n_i)$, while keeping the total number of samples fixed. In the original manuscript, we reported results on B8 (IR = 1) and ImB8 (IR = 5). The corresponding ACC values across different imbalance levels are as follows: 0.9812 (IR = 1), 0.9637 (IR = 5), 0.9463 (IR = 10),  and 0.9187 (IR = 100).
> > >
> > > We observe that the performance gradually decreases as the imbalance ratio increases, which is expected due to the increased difficulty of clustering under skewed distributions. Importantly, the degradation is smooth rather than abrupt, indicating that the proposed method remains stable across a wide range of imbalance levels.
> > >
> > > (2) We have added runtime comparisons on three datasets (CIFAR-10, MNIST, and Covtype) to better support the scalability claim. The runtime results (in seconds) are as follows:
> > >
> > > Method / CIFAR-10 / MNIST / Covtype
> > >
> > > K-means: 6.79 / 17.64 / 281.78
> > >
> > > KKM: 35.93 / 415.23 / OM
> > >
> > > CDKM: 7.91 / 28.12 / 335.05
> > >
> > > ERCAN: – / – / OM
> > >
> > > LCSOG: 15.37 / 28.44 / 328.82
> > >
> > > FCAG: 3.69 / 6.54 / 280.23
> > >
> > > ACLR: 9.15 / 14.36 / 284.54
> > >
> > > Ours: 4.19 / 6.82 / 271.18
> > >
> > > Our method achieves competitive runtime across all datasets and remains stable on large-scale data, while some sample-level methods (e.g., KKM and ERCAN) either run out of memory (OM) or require excessive computational time (“–”). These results further demonstrate the scalability advantage of our method in practice.
> > >
> > > We will incorporate these additional results and analyses into the revised version to further strengthen the empirical evaluation. We hope these clarifications address the reviewer’s concerns.

---

### Official Review · Reviewer_m6JF · 2026-03-11

**Soundness:** 3
**Presentation:** 3
**Significance:** 3
**Originality:** 3
**Overall Recommendation:** 5
**Confidence:** 5

**Summary:**

This paper introduces a Large-Scale Manifold Balanced Clustering framework that unifies manifold learning, anchor-based clustering, and class balance within a single optimization model. Comprehensive experiments on several benchmark datasets show superior performance compared to both sample-level and anchor-based clustering methods.

**Compliance With Llm Reviewing Policy:**

Affirmed.

**Final Justification:**

The authors' responses have addressed my concerns, and I tend to accept this paper.

**Key Questions For Authors:**

1. How does the proposed method relate to and differ from spectral clustering?
2. What clustering performance is achieved when optimizing Equation (12) without the Schatten-p norm term?
3. Could the authors enhance the figure quality and font size to improve readability?

**Limitations:**

Yes

**Strengths And Weaknesses:**

Strengths:
1. The proposed framework enables structural-label consistency by co-optimizing clustering assignments and the manifold representation, avoiding a two-stage procedure.
2. The theoretical analysis that relates maximizing the Schatten-p norm to balanced cluster sizes offers valuable understanding of the objective's function.
3. The algorithm is efficient in computation and memory usage relative to several sample-level manifold clustering baselines.
4. The method learns balanced clusters on Balanced-8 while maintaining the original distribution on Imbalanced-8, demonstrating that it does not artificially enforce balance.

Weaknesses:
1. The connection and distinctions between the proposed method and spectral clustering are not thoroughly explored.
2. The experiments do not include an ablation study to separately evaluate the contribution of the Schatten-p norm term.
3. The fonts used in the figures and captions in the experimental section are rather small, reducing readability.

---

> ### Author Rebuttal · Authors · 2026-03-31
>
> $\mathbf{Q1}$: Our model is similar in form to spectral clustering. If the distance matrix is replaced by a Laplacian matrix, it reduces to a spectral clustering framework. However, spectral clustering focuses on graph structure and performs clustering via eigen-decomposition of the Laplacian matrix, while our method operates in the anchor space and models pairwise relationships through an anchor-induced distance. In addition, our method integrates label consistency with manifold structure during optimization and explicitly model the relationship between structure preservation and clustering assignments.
>
> $\mathbf{Q2}$: Without the Schatten-p norm term, the ablation results are as follows:
>
> For AR: ACC = 0.3894, NMI = 0.6871, Purity = 0.4099
>
> For isolet: ACC = 0.6264, NMI = 0.7496, Purity = 0.6373
>
> For gisette: ACC = 0.8213, NMI = 0.3228, Purity = 0.8213
>
> For PEAL: ACC = 0.7729, NMI = 0.7368, Purity = 0.7729
>
> $\mathbf{Q3}$: Thank you for the suggestion. We will improve the figure quality and increase font sizes in the revision.

---

> > ### Author Rebuttal · Reviewer_m6JF · 2026-04-03
> >
> > The authors' responses have addressed my concerns.

---

> > > ### Author Response · Authors · 2026-04-03
> > >
> > > We sincerely thank the reviewer for the positive feedback and for acknowledging that the concerns have been fully addressed.

---

### Official Review · Reviewer_hU4D · 2026-03-12

**Soundness:** 3
**Presentation:** 4
**Significance:** 3
**Originality:** 3
**Overall Recommendation:** 4
**Confidence:** 5

**Summary:**

This paper proposes a manifold-balanced clustering method based on an anchor-induced distance. The proposed approach aims to ensure consistency between the manifold structure and clustering assignments by constructing the manifold structure from clustering labels and directly optimizing clustering labels on the learned manifold.

**Compliance With Llm Reviewing Policy:**

Affirmed.

**Key Questions For Authors:**

The following questions are intended to clarify several aspects of the methodology and experimental evaluation:

1. Baseline choice: could the authors elaborate on the motivation for selecting K-means as the representative baseline in Section 4.4, particularly given that other clustering algorithms in the comparison appear to perform more competitively?

2. Parameter selection: how were the key parameters of the proposed method selected across different datasets? Were they tuned separately for each dataset or fixed across experiments?

3. Performance variation: are there specific characteristics of datasets (e.g., size, feature dimensionality, or manifold structure) under which the proposed method tends to perform less effectively?

**Limitations:**

1. Baseline Selection for Comparison
Section 4.4 attempts to illustrate the effectiveness of the proposed method in achieving balanced cluster distribution by benchmarking against K-means. However, the results in Tables 2 and 3 show that K-means performed the worst among the benchmarked methods. It would be better if Figure 3 benchmarks the proposed method against the anchor-based method. This would have demonstrated that the recent state-of-the-art method does not achieve balanced clusters.

2. Discussion of Method Limitations
The manuscript would benefit from a discussion of the limitations of the proposed method. In particular, the results indicate that the proposed algorithm performs less favourably on some datasets (e.g., USPS and Covtype in terms of NMI and Purity). It would be helpful if the authors could discuss potential factors that may contribute to this behaviour, such as dataset characteristics, parameter sensitivity, or assumptions of the proposed approach.

3. Theorem 3.1: The author proposed the use of the Schatten-p norm to ensure uniform clustering distribution. Theorem 3.1 and its accompanying proof attempt to show that equation (3) leads to equal singular values of the label matrix. However, the optimality proof, especially with respect to the actual clustering objective, remains contested. For instance, (1) the theorem assumes that $\delta_1 = \delta_2 ... = \delta_K$, but this is not enforced during clustering (equation 12) - this can be interpreted as a characterization of the optimum, but the guarantee of reaching the optimum is not shown. (2) The proof only analyzes the Schatten-p norm of Z; it ignores the manifold structure $Tr(Z^TWZ)$. As a result, it is unclear how the trade-off between structure preservation and balanced distribution is resolved during optimization, or under what conditions the balance-inducing effect dominates - compare standard equation 2 with the refined objective in equation 12. And there are no empirical results to support this theorem 3.1

4. Experimental settings and reproducibility: The reported results are mainly attributed to the effectiveness of the proposed method. It would be helpful if the authors could provide detailed experimental settings, such as data preprocessing and parameter settings, which may also contribute to the observed performance.



Minor Concerns
- The evaluation metrics ACC, NMI, and Purity are used throughout the paper but are not explicitly defined. Providing a brief explanation or appropriate references for these metrics would improve clarity.

- Additional clarification regarding how the parameters of the proposed method were selected across datasets would improve the reproducibility of the experiments.

The concerns above point to areas that could benefit from further clarification or discussion, and some of these issues may be addressed with additional explanation from the authors.

**Strengths And Weaknesses:**

The manuscript is clearly written and generally well organized, making the proposed approach easy to follow.

The paper addresses a relevant problem in clustering by integrating manifold learning with clustering assignments.

The experimental evaluation considers multiple datasets and provides comparative analysis with existing clustering methods.

---

> ### Author Rebuttal · Authors · 2026-03-31
>
> $\mathbf{Q1}$: We thank the reviewer for the valuable suggestion. Section 4.4 mainly analyzes clustering distribution behavior under both balanced and imbalanced settings.  We use K-means as a classical distance-based clustering method as a reference for illustrating clustering distributions. We agree that including comparisons with anchor-based methods would further strengthen the persuasiveness of this part. We will incorporate such results in the revised manuscript to improve the analysis.
>
> $\mathbf{Q2}$: We thank the reviewer for the question. The parameter ranges are fixed across all datasets, while the optimal values are selected individually for each dataset. The proposed method involves three key parameters: the anchor ratio, the Schatten-p norm parameter $p$, and its weight $\lambda$. In our experiments, all parameters are tuned via grid search. Specifically, the $anchor ratio$ is selected from the range [0.1, 1.0], while $p$ and $\lambda$ are chosen from [1, 1.9] and [0.1, 1.0], respectively.
>
> $\mathbf{Q3}$: Thank you for this insightful comment. We observe that on USPS and Covtype, our method is slightly lower than some baselines on certain metrics . We believe this is mainly related to dataset characteristics and practical factors.
> For USPS, where clusters are simple and well-separated, we attribute the slight difference to the mild effect of the balancing regularization in our model. This regularization introduces a soft bias toward more balanced cluster assignments, which may slightly perturb the optimal partition when the intrinsic structure is already clear, leading to minor differences in metrics such as NMI.
> For Covtype, which is large-scale and exhibits complex data distribution, the underlying manifold structure may be less well-defined. Since our method relies on capturing meaningful structural information via manifold modeling, its effectiveness can be mildly affected when the data distribution is highly complex or does not admit a clear manifold structure, making it more challenging to fully capture the intrinsic structure.
>
> In addition, parameter selection (e.g., via grid search) may not always yield optimal configurations for every dataset, which can also contribute to small performance variations.
>
> Overall, these differences are minor, and our method still achieves competitive performance across most datasets. We will include a discussion of these limitations in the revised manuscript and explore more adaptive parameter and structure modeling strategies in future work.
>
> $\mathbf{L3}$: Thank you for your valuable comments.
> (1) Theorem 3.1 characterizes the optimal property of the Schatten-p norm, i.e., balanced clustering is achieved at the optimum. In our model, it is incorporated into Eq. (12) and jointly optimized with the structure term. Due to non-convexity, the global optimum is not guaranteed, but the regularization guides the solution toward more balanced assignments.
>
> (2) Eq. (12) jointly considers structure preservation and balance regularization, and the final solution is determined by their interplay. The parameter α adjusts their relative importance, while the actual balance depends on the data. Compared with Eq. (2), Eq. (12) introduces a bias toward balanced clustering via the Schatten-p norm while preserving structure.
>
> (3) Although Theorem 3.1 does not include the manifold term, its effect is supported empirically. Synthetic results show balanced clustering on balanced data and preserved distributions on imbalanced data, indicating the regularization works as intended without enforcing balance. Additional ablation studies will be included in the revision.
>
> $\mathbf{MC1}$: Thank you for the suggestion. We will include appropriate references for ACC, NMI, and Purity in the revised manuscript to improve clarity.

---

> > ### Author Rebuttal · Reviewer_hU4D · 2026-04-02
> >
> > My concerns are adequately addressed, but I do not want to change my score.

---

> > > ### Author Response · Authors · 2026-04-02
> > >
> > > We sincerely thank the reviewer for the positive feedback and for acknowledging that the concerns have been fully addressed.

---

### Official Review · Reviewer_GrSy · 2026-03-13

**Soundness:** 3
**Presentation:** 2
**Significance:** 3
**Originality:** 1
**Overall Recommendation:** 4
**Confidence:** 4

**Summary:**

The work discusses a relevant problem in unsupervised learning: scalable clustering that simultaneously captures manifold structure and avoids cluster imbalance. The authors attempt to assess a broad aspect of clustering quality by addressing three issues simultaneously: manifold structure preservation, clustering–structure consistency, and cluster balance.

**Compliance With Llm Reviewing Policy:**

Affirmed.

**Key Questions For Authors:**

- The manifold modeling uses $S=QQ^\top$ with Q derived from labels. This means the manifold is induced by labels, rather than learned from data. This may create a circular dependency: labels define manifold and then manifold defines labels. This resembles self-reinforcing clustering. Will this lead to local minima or trivial partitions?
- How are anchors selected?
- Why does performance decrease as the anchor rate increases in certain datasets, such as USPS?

**Limitations:**

yes

**Strengths And Weaknesses:**

## strengths
- The paper proposes a unified formulation linking manifold learning and clustering labels, where the manifold structure is explicitly induced by the label matrix (Eq.1–2). This perspective is conceptually interesting.
- Experiments cover eight datasets, including large-scale datasets such as MNIST and Covtype (up to ~581k samples).
- Large-scale clustering with balanced partitions is practically relevant in many real applications.

## weaknesses
- The paper claims scalability benefits, but runtime or memory comparisons are not reported. Only OM (out-of-memory) is shown in tables. Without runtime plots, scalability claims remain incomplete.
- There is no ablation evaluating: effect of Schatten-p regularization, effect of anchor-induced distance mapping and effect of manifold consistency term. This makes it difficult to understand which component contributes most.
- It would be more intuitive to visualize a comparison of clustering results for unbalanced datasets.

---

> ### Author Rebuttal · Authors · 2026-03-31
>
> $\mathbf{Q1}$: We thank the reviewer for this insightful question. The proposed model is non-convex, and thus, like most clustering methods, it may converge to a local optimum. This is a general property of iterative clustering algorithms rather than a consequence of the formulation. More importantly, the proposed method does not lead to trivial solutions. The Schatten-p norm regularization explicitly encourages balanced singular values of the label matrix, which prevents collapse to a single cluster or highly imbalanced partitions. In addition, although the manifold structure is constructed from the labels, the update of $\mathbf{Z}$ is jointly constrained by the data-dependent distance matrix $\mathbf{W}$. Therefore, the learning process is not purely self-reinforcing, and the labels cannot collapse arbitrarily.
>
> $\mathbf{Q2}$: Anchors are selected using the variance-based de-correlation anchor selection (VDA) strategy as described in “Tensorized Bipartite Graph Learning for Multi-View Clustering”. This method aims to select representative and diverse anchors by reducing redundancy and preserving the intrinsic data distribution. We will clarify this in the revised manuscript.
>
> $\mathbf{Q3}$: This phenomenon is mainly related to redundancy and over-concentration in the anchor space. For datasets such as USPS, which have compact structures, a small number of anchors is sufficient to capture the underlying manifold while reducing redundancy. Increasing the anchor ratio introduces redundant anchors, which degrades the quality of the learned structure and may lead to overfitting to local patterns, thereby resulting in decreased clustering performance.
>
> $\mathbf{W1}$: Thank you for the valuable comment. We have added runtime comparisons on three datasets (CIFAR-10, MNIST, and Covtype) to better support the scalability claim. The runtime results (in seconds) are as follows:
>
> Method / CIFAR-10 / MNIST / Covtype
>
> K-means: 6.79 / 17.64 / 281.78
>
> KKM: 35.93 / 415.23 / OM
>
> CDKM: 7.91 / 28.12 / 335.05
>
> ERCAN: – / – / OM
>
> LCSOG: 15.37 / 28.44 / 328.82
>
> FCAG: 3.69 / 6.54 / 280.23
>
> ACLR: 9.15 / 14.36 / 284.54
>
> Ours: 4.19 / 6.82 / 271.18
>
> Our method achieves competitive runtime across all datasets and remains stable on large-scale data, while some sample-level methods (e.g., KKM and ERCAN) either run out of memory (OM) or require excessive computational time (“–”). These results further demonstrate the scalability advantage of our method in practice.
>
> $\mathbf{W2}$: Thank you for the valuable suggestion. To better evaluate the contribution of each component, we will include the following ablation studies in the revised manuscript.
>
> (1) Without the Schatten-p norm term, the ablation results are as follows:
>
> For AR: ACC = 0.3894, NMI = 0.6871, Purity = 0.4099
>
> For isolet: ACC = 0.6264, NMI = 0.7496, Purity = 0.6373
>
> For gisette: ACC = 0.8213, NMI = 0.3228, Purity = 0.8213
>
> For USPS: ACC = 0.7729, NMI = 0.7368, Purity = 0.7729
>
> Without the Schatten-p norm term, the performance consistently drops across all datasets, indicating that the Schatten-p regularization contributes to improving clustering quality.
>
> (2) Replacing the anchor-induced distance with Euclidean anchor distance:
>
> For AR: ACC =0.3356, NMI =0.6202, Purity =0.3551
>
> For isolet: ACC =0.5625, NMI =0.6441, Purity =0.5660
>
> For gisette: ACC =0.6080, NMI =0.0455, Purity =0.6080
>
> For USPS: ACC =0.6516, NMI =0.5667, Purity =0.6627
>
> Replacing the anchor-induced distance with Euclidean anchor distance leads to the most significant performance degradation, suggesting that the anchor-induced distance plays a critical role in capturing meaningful structural relationships.
>
> (3) Replacing the manifold formulation with a standard distance-based objective:
>
> For AR: ACC =0.3590, NMI =0.6508, Purity =0.3782
>
> For isolet: ACC =0.5616, NMI =0.6745, Purity =0.5702
>
> For gisette: ACC =0.7471, NMI =0.1843, Purity =0.7471
>
> For USPS: ACC =0.7316, NMI =0.6389, Purity =0.7638
>
> The manifold consistency term also has a clear impact, showing its importance in preserving data structure.
>
> The ablation results show that removing any of the three components leads to a consistent performance drop across datasets, indicating that each component plays an essential and complementary role in the proposed framework.
>
> $\mathbf{W3}$: In the revised manuscript, we will include t-SNE visualizations to more intuitively demonstrate the effectiveness of the proposed method in handling imbalanced clustering.

---

### Decision · Program_Chairs · 2026-04-30

**Decision:**

Accept (regular)

**Comment:**

The reviewers appreciated the motivation, problem formulation and solution, as well as extensive experiments across multiple datasets, with varying sizes, numer of clusters, high dimensionality of features, and cluster balancedness, and relevant baselines.  Furthermore, ablation experiments during the rebuttal clarified the contribution, and run-time experiments confirmed scalability to large data. Enthusiasm for accepting the paper was moderate overall.

For the final version, comparison to clustering of landmark-based manifold learning should be added, e.g. ROSELAND, landmark Isomap. The imbalanced clustering experiment requested by reviewer SK6Y and performed in a very limited manner in the rebuttal should be extended to include comparison to baselines and added to the paper.

The AC recommends the writing be improved, explicitly including preliminaries and notations at the beginning of section 3 to clarify the dataset, the anchors, and the anchor selection approach (which was clarified during the rebuttal). Furthermore, the paper also doesn’t explain how cluster labels are assigned to the non-anchor datapoints, i.e., n-m remaining points, which is a major oversight (though it reviews anchor-based clustering in section 2.2). As two minor points: what is the JAFFE dataset referred to in the appendix? How is Balanced-8 / Imbalanced-8 constructed - what data is used?

The anchor ratio seems to have very different impact across datasets - i.e. for some lower ratio is better and for some higher ratio is better. It is not clear how to set this. Furthermore with very high ratio the computational advantage of the method is reduced. The authors should discuss this limitation explicitly in the paper and provide insight into how to set this parameter.